# Chasing the Wind: Background Flow Tracing for Wind Speed Forecasting

## Abstract

Wind energy is inherently intermittent and fluctuating, and the uncertainty in wind speed poses significant challenges for power system stability. Wind speed forecasting, particularly at wind farm sites, is crucial for balancing power generation and scheduling backup energy sources. Most existing studies rely solely on time-series forecasting methods, while ignoring the physical nature of wind as a spatiotemporal phenomenon primarily driven by atmospheric momentum advection. In this paper, we propose a framework that leverages the surrounding wind field as context to capture the wind dynamics. By explicitly modeling wind advection, our method traces atmospheric momentum transport, enabling the forecast of future trends. We further introduce a multi-modal dataset consisting of wind speed observations from multiple geographically distributed sites and their associated background wind field data. Experimental results demonstrate that our method achieves superior performance over traditional time-series forecasting models and state-of-the-art methods that leverage wind field information.

## 1 Introduction

Despite its importance as a renewable energy source, the uncertainty of wind speeds often causes large fluctuations in power output. Such drops or surges disrupt supply–demand balance. Wind-speed forecasting at wind farm sites is, therefore, critical for stable dispatch, effective reserve scheduling, and reliable grid integration (Holttinen, 2005; Möhrlen et al., 2022).

Previous studies define wind speed forecasting as a local time series problem (Wang et al., 2021; Hou et al., 2022). However, wind is not a local phenomenon; instead, it evolves dynamically and continuously, reflecting the behavior of large-scale atmospheric circulation systems. Therefore, the wind observed at a fixed location is often influenced not only by local conditions but also by air masses advected from other regions. Although recent studies have incorporated background field information, most studies simply append such information as additional variables, failing to model their underlying spatiotemporal evolution (Bone et al., 2018; Wang et al., 2025; Li et al., 2025). Recent methods use neural differential-equation to inject physics priors (Verma et al., 2024; Hettige et al., 2024; Tian et al., 2025). However, wind speed series forecasting concerns only local evolution, whereas these methods have to model the entire background field's evolution, incurring unacceptable computational cost.

Many atmospheric and oceanic variables evolve predominantly through advection(Vallis, 2017; Holton & Hakim, 2013). Because signals propagate along streamlines under advection, the down-stream evolution can be approximated by a time-shifted version of the upstream evolution. We theoretically prove that this time shift remains nearly unchanged over short term and can be approximated by a constant delay. This perspective represents the target's future as the superposition of (i) a constant-delayed upstream component, and (ii) a residual term accounting for non-advective effects and slight temporal distortions. We exploit this *constant-delay approximation property* by first identifying upstream regions whose earlier evolution matches the target's recent history, and then employing their recent history to forecast the target's future variations.

Motivated by the above analysis, we propose MoP, a novel framework designed to capture the underlying atmospheric **Mo**tion **P**atterns inherent in background meteorological fields. MoP comprises two main components: (1) Advection Backtracker which identifies potential upstream regions influencing the target location and generates a preliminary estimation; and (2) Residual Predictor

that corrects the preliminary estimation by exploring non-advective effects and temporal distortions. MoP employs a recurrent forecasting approach, advancing the forecasts in short lead time steps to maintain the local validity of the constant-delay approximation property. This design facilitates continual realignment with the latest background fields, minimizing the accumulation of temporal distortions. Finally, MoP uses the background field trend prediction as guidance and integrates it with historical wind-speed observations to produce the final wind speed prediction. Furthermore, the approach is applicable to other situations whose background fields evolve under advective transport, such as using cloud fields to assist irradiance prediction or leveraging radar echo patterns for precipitation forecasting (Bu et al., 2024; Agrawal et al., 2019; Lin et al., 2025).

Evaluating wind speed forecasting with background fields requires a dedicated dataset containing both gridded background fields and wind speed observations. However, to our knowledge, no such public dataset currently exists. To address this gap, we construct a new benchmark dataset that associates multi-station wind speed observations with their surrounding wind fields. The background fields are derived from ECMWF reanalysis products at a coarse spatial resolution, providing large-scale context to complement point-level observations. The benchmark covers both ultra-short-term (4h) and short-term (24h) forecasting tasks under in-domain and out-of-domain settings, enabling evaluation of accuracy and generalization. Results on the benchmark dataset demonstrate that our framework achieves substantial improvements over state-of-the-art methods in both in-domain and out-of-domain tests.

The main contributions of this paper can be summarized as follows:

- We provide a detailed analysis of the physical principles underlying atmospheric motion and derive a constant-delay advective approximation that relates downstream evolution to upstream signals through a near-constant delay. This theoretical foundation not only explains the design of MoP but also highlights its advantage in capturing advective dynamics for forecasting.

- We propose MoP, a novel framework that fully exploits background fields subject to advective transport to facilitates time series forecasting. MoP achieves superior performance in both ultra-short-term and short-term forecasting tasks, and generalizes well to unseen locations, demonstrating strong applicability in practice.

- We construct a new multi-modal dataset that associates observed wind speed data with their surrounding background wind field data. The dataset contains 4h and 24h forecasting tasks and includes in-domain and OOD splits to assess both accuracy and generalization.

## 2 RELATED WORK

**Traditional Time Series Forecasting.** In recent years, deep learning has been increasingly applied to time series forecasting and successfully adapted to meteorology, enabling accurate prediction of variables such as wind speed, temperature, and solar irradiance (Wang et al., 2021; Hou et al., 2022; Khouili et al., 2025). DLinear (Zeng et al., 2023) treats time series forecasting as a set of simple linear mapping problems, significantly reducing complexity while achieving competitive performance. FITS (Xu et al., 2023) formulates forecasting as a linear mapping in the Fourier domain. TimeMixer++ (Wang et al., 2024a) employs multi-resolution time imaging with dual-axis attention, while Time-MoE (Shi et al., 2024) leverages a mixture-of-experts structure for large-scale pretraining, demonstrating notable effectiveness. Among Transformer-based architectures, Informer (Zhou et al., 2021) introduces sparse attention to improve scalability for long sequences, while Autoformer (Wu et al., 2021) enhances its capability to model long-term dependency through auto-correlation calculation and trend–seasonality decomposition. FEDformer (Zhou et al., 2022b) further integrates Fourier transforms to improve periodic pattern learning in the frequency domain. PatchTST (Nie et al., 2022) introduces a patch-based transformer architecture for time series forecasting, achieving state-of-the-art accuracy. However, these methods rely solely on historical observation series, whereas meteorological variables evolve across spatiotemporal fields rather than at isolated points. As a result, conventional time-series methods often struggle in meteorological forecasting.

**Multi-Modal Time Series Forecasting.** To address the limitations of traditional methods, a growing number of studies have incorporated additional spatiotemporal information to assist time series forecasting (Bone et al., 2018; Wang et al., 2025; Li et al., 2025). CrossViViT (Boussif et al., 2023)

integrates satellite cloud imagery with historical solar irradiance data to predict future solar irradiance. Built upon CrossViViT, Fusion-SF (Ma et al., 2024) employs a vector quantization framework to harmonize representations across varying information densities, enabling effective information integration while mitigating overfitting. WindDragon (Keisler & Le Naour, 2025) transforms numerical weather prediction (NWP) wind fields into image sequences and applies visual backbones to capture spatial patterns for regional wind power forecasting. These methods typically use background fields as static context, lacking deeper exploration and utilization of underlying evolution mechanisms of the background fields. In contrast, we explicitly model the advective dynamics, identifying upstream regions with similar trends and following their propagation to the site.

**Physics-guided Advection Modeling.** A growing number of studies integrate physics priors into methods for atmospheric environment prediction (Chen et al., 2025; 2024). ClimODE (Verma et al., 2024) converts the PDEs that describe atmospheric motion into an ODE system via the Method of Lines (MOL), and then trains a neural ODE to predict the evolution of the gridded fields. AirPhyHet (Hettige et al., 2024) represents advection and diffusion with differential-equation networks and uses a graph structure to inject physics priors while capturing spatiotemporal relations in air-quality data. Air-DualODE (Tian et al., 2025) couples a physics ODE branch with a data-driven branch to model open-system pollutant dynamics. While effective for field propagation, these approaches operate on the full background grid and typically require numerical integration or multi-step updates, which is computationally heavy, making them ill-suited for time series forecasting. In contrast, our work exploits the constant-delay property to identify upstream regions and learns a residual term to correct deviations from advection, thereby avoiding computationally prohibitive full-grid processing in high resolution data.

## 3 METHODOLOGY

### 3.1 PHYSICAL MOTIVATION

The evolution of atmospheric and oceanic variables is primarily governed by two processes: advection, which transports properties along the flow, and diffusion, which smooths gradients through molecular or turbulent diffusion. Their relative importance can be measured by the Péclet number $Pe$. $Pe > 1$ indicates that advection dominates over diffusion. For typical atmospheric conditions, $Pe$ commonly falls in the range $10^1$–$10^6$ or higher (Tennekes & Lumley, 1972; Holton & Hakim, 2013; Vallis, 2017), implying that the variable evolution is controlled mainly by advection. The calculation of Pe is detailed in Appendix A. Inspired by this, we consider advection as the primary process that governs the evolution of the background field. The advection of a scalar field $q(\mathbf{x}, t)$ under a velocity field $v(\mathbf{x}, t)$ is governed by the advection equation:

$$\frac{\partial q}{\partial t} + v \cdot \nabla q = 0. \tag{1}$$

This equation describes the idealized scenario in which the scalar quantity is passively transported along the flow without diffusion or external sources, maintaining its value along trajectories in the flow field. From this property, we derive the following theorem:

**Theorem 1: Constant-Delay Approximation for Advected Signals.** *Consider a spatial location $(x_1, y_1)$ lying on the streamline of a flow, with the streamline topology remaining unchanged. For a small time delay $\Delta t$, there exists an upstream point $(x_2, y_2)$ such that the scalar quantity q at $(x_1, y_1)$ can be represented as its value at $(x_2, y_2)$ with a constant delay $\Delta t$ plus a small residual $\epsilon(t)$. That is,*

$$q(x_1, y_1, t) = q(x_2, y_2, t - \Delta t) + \epsilon(t). \tag{2}$$

In practice, the invariance assumption on streamline topology is generally valid, as large-scale flow structures rarely undergo abrupt topological changes over short horizons, and their variations typically occur on the scale of weeks. This theorem provides an inductive bias: a downstream signal can be approximated by a time-shifted upstream signal plus a residual error. These residual arises from two sources: temporal distortions induced by fluctuations in advective speed, and non-advective effects such as friction. This motivates our design of the **Advection Backtracker**, which identifies upstream regions and estimates advective trends, and the **Residual Predictor**, which corrects the residual errors. For a full derivation of the constant delay approximation for advected signals, please refer to Appendix B.

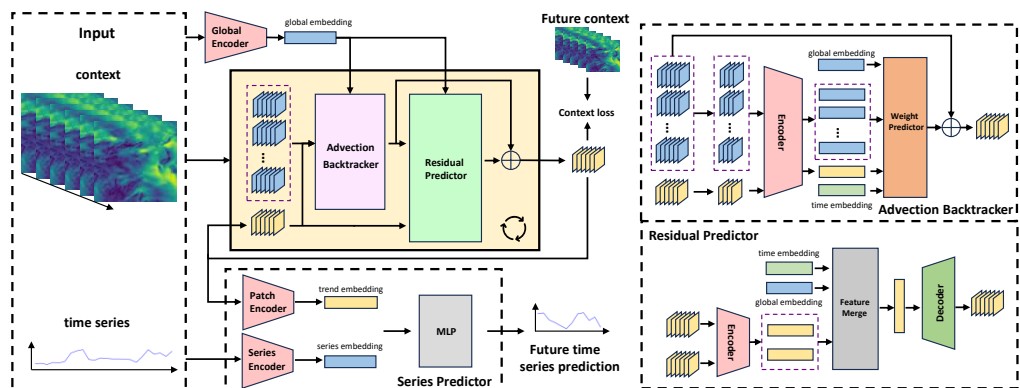

Figure 1: **The architecture of the MoP framework.** It captures wind propagation through two modules: (1) Advection Backtracker traces upstream regions to yield advective estimates; (2) Residual Predictor corrects the residuals. A recurrent mechanism enables long-horizon advective estimates. Finally, Series Predictor combines advective estimates and historical observations to generate the prediction.

## 3.2 OVERVIEW

We propose a novel time series forecasting framework named MoP, which explicitly models the advective momentum transport in background fields. MoP consists of two key modules: **Advection Backtracker** identifies upstream regions and estimates advective trends and **Residual Predictor** corrects non-advective effects and temporal distortions. In addition, a lightweight module **Series Predictor** combines the corrected advective trends with historical time series observations to produce the final forecast.

Formally, given background field data $F \in \mathbb{R}^{C_{\text{bf}} \times T_{\text{bf}} \times H \times W}$ and historical time series observations $S \in \mathbb{R}^{C_{\text{ts}} \times T_{\text{ts}}}$, MoP generates wind speed forecast $\hat{y} \in \mathbb{R}^{C_{\text{ts}} \times T_h}$. Here, $C_{\text{bf}}$ and $C_{\text{ts}}$ denote the numbers of channels in the background field and the time series, $T_{\text{bf}}$ and $T_{\text{ts}}$ denote their temporal lengths, $H$ and $W$ are the spatial dimensions, and $T_h$ is the forecast horizon.

Given background field data $F$, we first extract a global context vector $g = \text{Encoder}_{\text{global}}(F)$ to summarize the global circulation patterns for subsequent modules. The background field data are then partitioned into spatiotemporal patches $P_{\text{candidates}} = \{p_1, \ldots, p_N\}$, which serve as candidate regions for search. Each patch $p_i \in \mathbb{R}^{C_{\text{bf}} \times T_{\text{bf}} \times H_p \times W_p}$ represents the local spatiotemporal evolution within a region of size $H_p \times W_p$ over $T_{\text{bf}}$ time steps. The patch centered on the target site serves as the initial query patch $p_{\text{query}}^{t=0}$ for the subsequent upstream region search.

With a fixed lead time $\Delta t$, the model rolls out for $K$ times (i.e., the horizon $T_h = K\Delta t$). At each step $t = i$, Advection Backtracker matches the current query $p_{\text{query}}^{t=i}$ against $P_{\text{candidates}}$ to identify upstream regions whose evolution leads the target by $\Delta t$, yielding a coarse advective estimate $p_{\text{Adv}}^{t=i}$. To account for non-advective effects and slight temporal distortions, the Residual Predictor infers a correction $\Delta p^{t=i}$ from $\{p_{\text{query}}^{t=i}, p_{\text{Adv}}^{t=i}\}$. The query is then updated by

$$p_{\text{query}}^{t=i+1} = p_{\text{Adv}}^{t=i} + \Delta p^{t=i}. \tag{3}$$

The updated query $p_{\text{query}}^{t=i+1}$ advances the forecast by lead time $\Delta t$, moving the horizon from $i\Delta t$ to $(i+1)\Delta t$. Repeating this update yields the prediction sequence $P_{\text{predictions}} = \{p_{\text{query}}^{t=1}, \ldots, p_{\text{query}}^{t=K}\}$.

Finally, Series Predictor fuses the observed historical series $S$ with the target region's evolution predictions $P_{\text{predictions}}$ to generate wind speed forecast $\hat{y}$.

## 3.3 ADVECTION BACKTRACKER

The Advection Backtracker module builds on Theorem 1: For a small time $\Delta t$, there always exists an upstream location such that the target location's evolution is well-approximated by its $\Delta t$-delayed

evolution. Specifically, we account for the constant delay $\Delta t$ and align the first $T_{\text{bf}} - \Delta t$ frames of the candidate patches $P_{\text{candidates}}$ with the target region's recent $T_{\text{bf}} - \Delta t$ frames. These upstream regions' recent $\Delta t$ frames then provides a coarse-estimate of the target's future evolution. To ensure the validity of the theorem, $\Delta t$ must be small. In practice, we choose $\Delta t = $ 1-4 h, with the exact value determined by the task setup.

The aligned patch segments are then passed through a shared patch encoder, yielding embeddings that capture their local trends: $x_{\text{query}}$ for the query patch and $\{x_1, ..., x_N\}$ for the candidate patches. Mapping all the patches into the unified space can reduce the impact of discrepancies and enable more accurate similarity matching between the query and candidates. Additionally, we concatenate the global context $g$ and temporal embedding $t$ with the query embedding $x_{\text{query}}$, then apply a linear mapping to obtain the final query vector $\hat{x}_{\text{query}}$. In this way, candidates with similar local trends can be better distinguished by the global and temporal information, enhancing the robustness of upstream region matching.

To retrieve relevant upstream region patches, we employ retrieval–reasoning blocks: cross-attention acts as a retrieval operator, and a feed-forward neural network (FFN) performs the reasoning step that refines the retrieved signal. Stacking these blocks gradually sharpens the query's focus on the most relevant upstream region patches. Denote the cross-attention operator by $\mathcal{A}(\cdot, \cdot)$ over the candidate set $X = [x_1; \ldots; x_N]$, the $\ell$-th layer updates the query state as:

$$z^{(\ell+1)} \;=\; \text{FFN}\big(z^{(\ell)} \;+\; \mathcal{A}\big(z^{(\ell)}, X\big)\big), \;\; \ell = 0, \ldots, L-1, \;\; z^{(0)} = \hat{x}_{\text{query}}. \tag{4}$$

After $L$ layers, the final query $z^{(L)}$ is mapped to similarity scores for all candidates, followed by softmax normalization to yield the final weight distribution $\{w_1, ..., w_N\}$. The final prediction is computed as a weighted sum of the original candidate patches:

$$p_{\text{adv}}^{t=i} = \sum_{k=1}^{N} w_k \cdot p_k. \tag{5}$$

The patch $p_{\text{adv}}^{t=i} \in \mathbb{R}^{C_{\text{bf}} \times T_{\text{bf}} \times H_p \times W_p}$ serves as an coarse approximation of the target's evolution from $(i+1)\Delta t$ to $(i+1)\Delta t + T_{bf}$.

## 3.4 RESIDUAL PREDICTOR

While the advection-based estimate captures the dominant momentum transport, the actual evolution often deviates due to non-advective effects and temporal distortions. Residual Predictor refines the coarse predictions from Advection Backtracker by explicitly modeling the residual.

This module corrects the advective estimate using four inputs: the historical query patch $p_{\text{query}}^{t=i}$, the advective prediction $p_{\text{Adv}}^{t=i}$, the global context $g$, and the temporal embedding $t$. Each input is embedded into a $d$-dimensional token and stacked as a short sequence $Z^{(0)} \in \mathbb{R}^{4 \times d}$. A self-attention fusion block with $L$ layers allows information exchange among the four inputs and refines the representation. Denote $\text{MSA}(\cdot)$ as multi-head self-attention and $\text{FFN}(\cdot)$ as a feed-forward neural network, we have

$$Z^{(\ell+1)} = \text{FFN}\big(Z^{(\ell)} + \text{MSA}(Z^{(\ell)})\big), \;\; \ell = 0, \ldots, L-1. \tag{6}$$

Subsequently, we apply a lightweight decoder to predict the residual:

$$\Delta p^{t=i} = \text{Dec}(Z^L) \in \mathbb{R}^{C_{\text{bf}} \times T_{bf} \times H_p \times W_p}. \tag{7}$$

## 3.5 SERIES PREDICTOR

The Series Predictor integrates two complementary signals to produce the forecast. The advective predictions based on background field $P_{\text{predictions}}$ provide reliable trend information with coarse spatiotemporal resolution, whereas the historical time series $S$ supplies fine-grained local detail with limited foresight. To exploit both sources, we employ a time-series encoder to extract local temporal dynamics from the historical sequence, and a patch encoder to capture the trending information from the advective predictions. These representations are concatenated and then passed through a linear projection to obtain the final prediction $\hat{y}$.

### 3.6 Loss Description

During training, we employ two losses: a primary loss on the final time-series forecast and an auxiliary loss on the query-patch predictions. This design encourages the MoP to learn both the local temporal dynamics and the spatiotemporal patterns within the background field.

First, at each iteration $t = i$, the predicted query patch $p_{\text{query}}^{t=i+1}$ from the Advection Backtracker and Residual Predictor modules is compared with the ground-truth sequence $p_{\text{ground-truth}}$ using Mean Absolute Error (MAE), ensuring MoP is consistently supervised to follow the background field evolution:

$$\mathcal{L}_{\text{patch}}^{t=i} = \left\| p_{\text{query}}^{t=i+1} - p_{\text{ground-truth}}[(i+1)\Delta t : (i+1)\Delta t + T_{\text{bf}}] \right\|_1 \tag{8}$$

Second, for the final time series prediction $\hat{y}$, we minimize the MAE against the observed sequence $y_{\text{ground-truth}}$:

$$\mathcal{L}_{\text{series}} = \left\| \hat{y} - y_{\text{ground-truth}} \right\|_1 \tag{9}$$

The total loss is defined as a weighted combination of the two terms:

$$\mathcal{L}_{\text{total}} = \mathcal{L}_{\text{series}} + \lambda \cdot \sum_i \mathcal{L}_{\text{patch}}^{t=i} \tag{10}$$

Here, $\lambda$ is a hyperparameter that adjusts the importance of query-patch prediction loss.

## 4 Dataset

This section provides an overview of the proposed multi-modal wind speed dataset (MMWS), which will be released publicly in the future. For data security purpose, the dataset has been anonymized by removing all geographical coordinates, timestamps, and identifiable metadata to prevent any potential identification of the data sources.

**Historical time series** are collected from four anemometer towers located at distinct geographic sites, each recording wind speed at a fixed height with a temporal resolution of 15 minutes. **Background wind fields** are obtained from ECMWF reanalysis products (Soci et al., 2024), representing large-scale wind conditions over a $H \times W$ grid centered around each tower location. The background fields have a spatial resolution of $0.25°$ and a temporal resolution of 1 hour. Both datasets are retained at their native resolutions without any form of temporal alignment or interpolation.

We define each sample as a pair $(S, F)$, where $S \in \mathbb{R}^{1 \times T_{ts}}$ denotes the historical wind speed sequence at the target location, and $F \in \mathbb{R}^{1 \times T_{bf} \times H \times W}$ represents the corresponding historical background wind speed field. $T_{ts} = 16$ and $T_{bf} = 4$ (corresponding to 4 hours) are used for the ultra-short-term forecasting task, while $T_{ts} = 96$ and $T_{bf} = 24$ (corresponding to 24 hours) are used for the short-term forecasting task. The forecasting horizons are $T_h = 16$ for 4-hour-ahead forecasting and $T_h = 96$ for 24-hour-ahead forecasting, respectively.

We use data from three out of four anemometer towers and perform a chronological split into training/validation/test sets at a 70%/15%/15% ratio to perform in-domain evaluation. We discard all sample that crosses the split boundary to prevent information leakage. For out-of-domain evaluation, we use the remaining tower as an unseen test dataset to assess cross-location transferability. The in-domain dataset contains approximately 22,000 samples for both ultra-short-term and short-term forecasting task, while the out-of-domain dataset contains about 4,000 samples for each task.

## 5 Experiment

### 5.1 Baselines

The baselines comprise two categories of methods. **Traditional time-series forecasting methods** include Crossformer (Zhang & Yan, 2023), PatchTST (Nie et al., 2022), FiLM (Zhou et al., 2022a), DLinear (Zeng et al., 2023), LightTS (Zhang et al., 2022), FITS (Xu et al., 2023), PatchMLP (Tang & Zhang, 2025), PDF (Dai et al., 2024) and TimeMixer (Wang et al., 2024b). These methods rely solely on historical observations without incorporating auxiliary context. **Multi-modal time-series forecasting methods** include CrossViViT (Boussif et al., 2023) and FusionSF (Ma et al., 2024), which use background field information as additional features.

Table 1: Overall forecasting results on the proposed benchmark. MAE / RMSE are reported for ultra-short-term (4h) and short-term (24h) forecasting under both in-domain and out-of-domain (OOD) settings. Best results are in **bold**, and the second best are underlined.

| Model | Ultra-Short-Term Forecasting (4h) | | | | Short-Term Forecasting (24h) | | | |
|---|---|---|---|---|---|---|---|---|
| | In-domain | | OOD | | In-domain | | OOD | |
| | MAE | RMSE | MAE | RMSE | MAE | RMSE | MAE | RMSE |
| Crossformer | 0.327 | 0.468 | 0.305 | 0.437 | 0.584 | 0.786 | 0.523 | 0.707 |
| PatchTST | 0.340 | 0.488 | 0.320 | 0.458 | 0.648 | 0.888 | 0.606 | 0.823 |
| FiLM | 0.340 | 0.488 | 0.320 | 0.457 | 0.654 | 0.898 | 0.576 | 0.787 |
| DLinear | 0.328 | 0.471 | 0.307 | 0.439 | 0.587 | 0.787 | 0.534 | 0.717 |
| LightTS | 0.327 | 0.469 | 0.306 | 0.438 | 0.589 | 0.792 | 0.532 | 0.718 |
| FITS | 0.340 | 0.488 | 0.320 | 0.457 | 0.655 | 0.898 | 0.575 | 0.785 |
| PatchMLP | 0.329 | 0.471 | 0.307 | 0.439 | 0.587 | 0.794 | 0.530 | 0.722 |
| PDF | 0.339 | 0.487 | 0.321 | 0.459 | 0.654 | 0.900 | 0.581 | 0.792 |
| TimeMixer | 0.327 | 0.470 | 0.306 | 0.438 | 0.586 | 0.786 | 0.527 | 0.711 |
| CrossViViT | 0.313 | 0.439 | 0.303 | 0.424 | 0.586 | 0.788 | 0.532 | 0.719 |
| FusionSF | 0.311 | 0.434 | 0.304 | 0.424 | 0.582 | 0.777 | 0.545 | 0.727 |
| **MoP (Ours)** | **0.304** | **0.423** | **0.292** | **0.407** | **0.536** | **0.718** | **0.493** | **0.662** |

We evaluate all methods on both forecasting horizons (4-hour and 24-hour) under two settings: in-domain and out-of-domain. Forecast accuracy is reported with mean absolute error (MAE) and root mean squared error (RMSE); lower values indicate better performance. For each setting, all baselines are trained with the same input lengths and forecasting horizons as MoP. Besides, models are provided with identical historical series inputs and, for context-aware methods, identical background field data to guarantee a fair comparison. For baselines, we used the default hyperparameters from the original papers and official code, modifying only the input and output dimensionalities to align with our task specification. Appendix C details the hyperparameter settings for MoP.

## 5.2 PERFORMANCE AND ANALYSIS

**Ultra-Short-Term Forecasting (4h).** In the in-domain task, multi-modal methods achieve significant improvements over traditional time-series forecasting methods. While compared with the best time-series baseline Crossformer, FusionSF reduces MAE by 4.9% and RMSE by 7.3%. This demonstrates that integration of background field information provides valuable signals for short-horizon forecasting. Compared with FusionSF, MoP achieves a further 2.2% lower MAE and 2.5% lower RMSE. The improvement is less pronounced than in the 24h forecasting task. This is mainly because the 4-hour forecasting horizon involves limited evolution, for which static feature extraction modules are already sufficient to capture the relevant dynamics. Even so, modeling advective transport explicitly still brings measurable benefits. Within the out-of-domain task, multi-modal methods performs closely to the traditional time-series baselines. This phenomenon indicates that contextual patterns differ across regions, limiting the transferability of static context feature extraction. In contrast, MoP delivers the best performance, reducing MAE by 3.6% and RMSE by 4.0% compared to the best baseline, CrossViViT. This demonstrates that explicitly modeling advective transport is less constrained by local idiosyncrasies and exhibits strong transferability across locations.

**Short-Term Forecasting (24h).** For longer forecast horizons, the best multi-modal method yields only marginal improvements over the strongest time-series baseline: 0.3% in MAE and 1.1% in RMSE. The small gain indicates that multi-modal methods struggle to capture the long-horizon evolution of the background context. Our proposed MoP further improves the accuracy. Compared with the best baseline FusionSF, MoP reduces MAE by 7.9% and RMSE by 7.7%. In the out-of-domain task, MoP remains the top performer, significantly outperforming all baselines. By explicitly modeling advection and correcting residual effects, MoP is able to track changes in the background fields over longer horizons, thus achieving superior accuracy. Its advection-aware design grounds predictions in physically guided cues rather than local idiosyncrasies, ensuring more reliable generalization under distribution shifts.

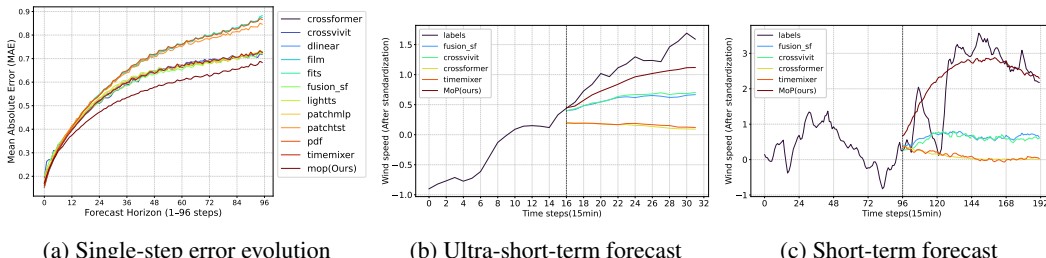

(a) Single-step error evolution     (b) Ultra-short-term forecast     (c) Short-term forecast

Figure 2: Visualization of forecast results. (a) Single-step forecast error evolution (MAE) under short-term forecasting settings. (b) An example from the ultra-short-term (4 h) forecasting task. (c) An example from the short-term (24 h) forecasting task.

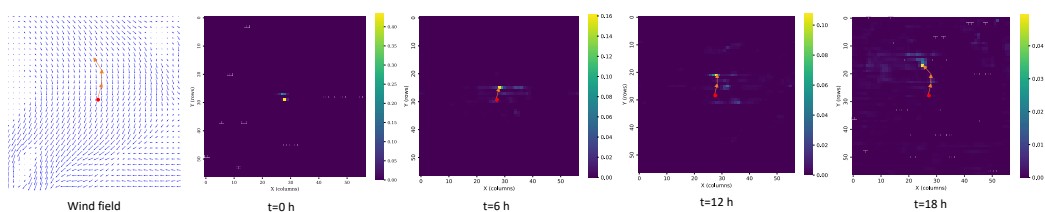

Figure 3: From left to right: the wind field and heatmaps of weight distribution for candidate upstream regions at $t = 0, 6, 12, 18\,\text{h}$. In the wind field, blue arrow direction indicates wind direction, and arrow length indicates wind speed magnitude. The red dot marks the target station, and orange arrows indicate the shift of the most influential upstream region over time.

## 5.3 BEHAVIOR ANALYSIS AND VISUALIZATION

**Single-step Forecast Error Evolution.** In the 24h forecasting task, we calculate the MAE for each prediction step from 1 to 96 and report the results in Figure 2a. At early steps, the curves are close, indicating similar short-range accuracy across methods. As the steps grows, MAE increases for all methods and the curves separate gradually. The multi-modal baselines and the time-series baselines (except for several methods that are not well-suited to this task) exhibit similar performance across most forecast steps, suggesting that directly extracting features from the context contributes little at longer horizon task. By contrast, MoP consistently achieves the lowest errors across the entire range, with its performance margin widening from early to mid-forecast steps and remaining stable through the mid-to-late steps. This results indicate that explicitly modeling advective transport is effective over long range.

**Visualization of Forecasts.** We present examples for the 4-hour and 24-hour forecasting setting. In the Figure 2b, an example from the ultra-short-term (4 h) forecasting task, traditional time-series baselines effectively flatline, producing near-constant forecasts with little upward or downward movement. This phenomenon stems from the models' inability to anticipate trend shifts, causing predictions to fluctuate around the most recent input. By contrast, multi-modal baselines show some improvement. They capture the rising trend but underestimate its magnitude. In Figure. 2c, an example from the short-term (24 h) forecasting task, as the forecast horizon increases, the future evolution becomes harder to anticipate, and most models exhibit flat, trend-agnostic predictions. In contrast, MoP captures both the timing and magnitude of changes, yielding the most accurate trajectory. By explicitly tracing advection in the background field, MoP captures wind evolution and predicts the timing and magnitude of future changes accurately, especially for longer horizons.

**Weight Visualization.** To interpret the model's internal mechanism, we visualize the weight distribution calculated by the Advection Backtracker module. Figure 3 shows four heatmaps of weight distribution from the iterative loop at $t = 0, 6, 12, 18\,\text{h}$. The visualization reveals a spatial-temporal progression: as the forecast steps advance, MoP shifts its weight distribution upstream, effectively following the pathways of the background flow. This upstream-tracking behavior supports our phys-

Table 2: Ablation on the proposed benchmark. MAE / RMSE are reported for 4h and 24h.

| Model | Ultra-Short-Term (4h) | | Short-Term (24h) | |
|---|---|---|---|---|
| | MAE | RMSE | MAE | RMSE |
| MoP (w/o AB and RP) | 0.313 | 0.447 | 0.584 | 0.791 |
| MoP (w/o RP) | 0.306 | 0.427 | 0.547 | 0.741 |
| **MoP** | **0.304** | **0.423** | **0.536** | **0.718** |

Table 3: Ablation on the lead time $\Delta t$. The lower MAE/RMSE reflects better performance.

| $\Delta t =$ | 1h | 2h | 3h | 4h | 6h | 8h | 12h |
|---|---|---|---|---|---|---|---|
| MAE | 0.559 | 0.549 | **0.536** | 0.546 | 0.551 | 0.554 | 0.558 |
| RMSE | 0.742 | 0.728 | **0.718** | 0.719 | 0.721 | 0.724 | 0.735 |

ical motivation and explains MoP's strong performance in long forecast horizon, as it can selectively exploit the relevant upstream information at each step.

We also observe that the weight distribution becomes progressively less concentrated as time advances: its spatial distribution becomes more diffuse, with a reduction in peak magnitude. This phenomenon stems from error accumulation during the recurrent rollout: at each step, small mismatches in the advective estimate and the learned residual are fed back into the query, progressively degrading the upstream matching accuracy and making subsequent identification harder.

### 5.4 ABLATION

**Module Ablation.** We conduct ablation experiments to assess the contributions of the Advection Backtracker (AB) and Residual Predictor (RP) modules, with results reported in Table 2. Our analysis focuses on the 24h forecasting task, since the 4h task is relatively easier and yields close results. In the MoP model without AB and RP, the context is processed through a global encoder and fed into the Series Predictor directly. Without explicitly modeling the physical mechanisms of the background field, the model degenerates into a generic multi-modal method, and performance drops to the level of traditional baselines. This shows the advantage of explicitly modeling the physical mechanisms in the background field. Retaining AB improves performance significantly but still slightly lags behind the full model. This demonstrates that background field evolution is mainly governed by advective motion. Adding RP yields further improvements, as it addresses residual effects such as temporal distortions and non-advective processes that contribute to time series forecasting.

**Lead time $\Delta t$.** The choice of the lead time $\Delta t$ balances two opposing effects. On one hand, a smaller $\Delta t$ increases the number of recurrent steps, leading to error propagation and accumulation, which progressively degrades upstream matching accuracy. On the other hand, the constant-delay approximation is accurate at small $\Delta t$. As $\Delta t$ increases, temporal distortions increase and upstream region search becomes less reliable. Accordingly, we conduct an analysis on the 24h task at $\Delta t = \{1, 2, 3, 4, 6, 8, 12\}$ hours (Table 3). The best results occur at $\Delta t = 3$ h. Our analysis is consistent with the aforementioned trade-off: small $\Delta t$ suffers from error accumulation over iterations, whereas large $\Delta t$ weakens the constant-delay assumption and reduces matching fidelity.

## 6 CONCLUSION

We introduced MoP, a novel framework that produces wind speed forecasting by leveraging the background field information. MoP explicitly models advective transport to predict the future evolution of the target region and combines historical sequences to generate highly accurate forecast results. Across ultra-short and short forecasting tasks, in both in-domain and OOD settings, MoP consistently outperforms time-series and multi-modal baselines. Single-step curves and weight maps show that MoP follows flow-consistent pathways and anticipates ramps more reliably. Future work will address the accumulation of errors at longer steps and extend the approach to other advected variables.

## 7 LLM USAGE DISCLOSURE

ChatGPT was used to assist with proofreading. All content was reviewed and verified by the authors.

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

## A  PÉCLET NUMBER FOR THE TASK

**Definition.**  Following Tennekes & Lumley (1972); Holton & Hakim (2013), we quantify advection–diffusion balance by the Péclet number

$$\mathrm{Pe} \;=\; \frac{U\,L}{\kappa},\tag{11}$$

where $U$ is a characteristic speed, $L$ is a characteristic length, and $\kappa$ is a diffusivity.

**Ranges of characteristic speed and length.**  In atmospheric dynamics, wind field evolution is typically described over meso- to synoptic-scale horizontal, which spans tens to hundreds of kilometers, and sometimes larger. Therefore, we consider horizontal transport with

$$U \in [1,\ 20]\ \mathrm{m\,/s}, \qquad L \in [1\times 10^5,\ 1\times 10^6]\ \mathrm{m}.$$

**Ranges of diffusivities.**  *Molecular diffusivity* stems from random thermal motion of molecules and is a property of the fluid itself. For air, it is extremely small (Marrero & Mason, 1972):

$$\kappa_{\mathrm{mol}} \approx 0.00001\ \mathrm{m^2/s}.\tag{12}$$

*Turbulent diffusivity* parameterizes the effective mixing by turbulent eddies. It depends on shear, stability, and mixing length, and is much larger than molecular values. For the atmospheric conditions (Blais et al., 1975; Dejesusparada et al., 1981) , we adopt a representative range:

$$K_H \in [10^1,\ 10^4]\ \mathrm{m^2/s}.\tag{13}$$

**Péclet numbers.**  *(i) Using molecular diffusivity:*

$$\mathrm{Pe}_{\mathrm{mol}} \in \big[\frac{1\times 10^5}{0.00001},\ \frac{20\times 10^6}{0.00001}\big] = \big[\,10^{10},\ 2\times 10^{12}\,\big].\tag{14}$$

*(ii) Using turbulent diffusivity:*

$$\mathrm{Pe}_H \in \big[\frac{1\times 10^5}{10^4},\ \frac{20\times 10^6}{10^1},\big] = \big[\,10^1,\ 2\times 10^6\,\big].\tag{15}$$

**Implication.**  Across these ranges, Pe is much greater than 1 for both molecular and turbulent diffusions. Therefore, the background wind field evolution in this work is dominated by *advection*. Diffusion plays a secondary role in smoothing. This analysis justifies our design choice to treat advection as the primary process in background field evolution.

## B  PHASE SHIFT PROPERTY UNDER TIME-VARYING VELOCITY FIELDS

Consider a two-dimensional scalar field $q(x,y,t)$ advected by a velocity field $\mathbf{v}(x,y,t) = (u(x,y,t), v(x,y,t))$. The advection equation is given by:

$$\frac{\partial q}{\partial t} + u(x,y,t)\frac{\partial q}{\partial x} + v(x,y,t)\frac{\partial q}{\partial y} = 0.\tag{16}$$

We seek to understand how the time series of $q$ evolves at different spatial locations connected by the flow. For this purpose, we analyze the motion of fluid particles from a Lagrangian perspective.

**Theorem A.1 (Scalar Invariance Along Lagrangian Trajectories)** *Let $(x(t), y(t))$ denote the trajectory of a fluid parcel starting from $(x_0, y_0)$ at time $t = 0$. Its path satisfies the ordinary differential equations:*

$$\frac{dx}{dt} = u(x(t), y(t), t), \quad x(0) = x_0,\tag{17}$$

$$\frac{dy}{dt} = v(x(t), y(t), t), \quad y(0) = y_0.\tag{18}$$

*Then, the scalar quantity $q$ remains invariant along the trajectory:*

$$\frac{d}{dt}q(x(t), y(t), t) = 0.\tag{19}$$

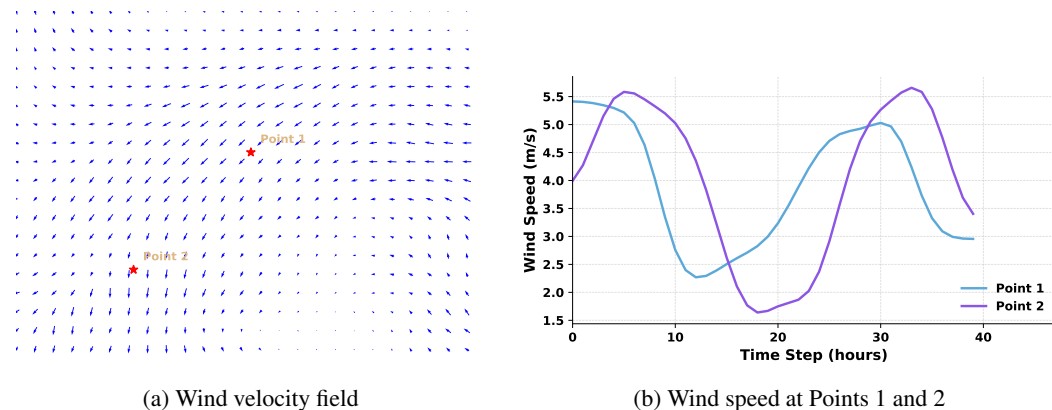

(a) Wind velocity field        (b) Wind speed at Points 1 and 2

Figure 4: **Theorem sample.** (a) Wind field visualization. Arrow direction indicates wind direction, and arrow length indicates wind speed magnitude. (b) Comparison of wind speeds at two points selected from the wind field (as shown in (a)). The blue curve represents the wind speed at point 1, and the purple curve represents point 2.

*Proof.* Applying the chain rule, the derivative along the trajectory is:

$$\frac{d}{dt}q(x(t), y(t), t) = \frac{\partial q}{\partial x} \cdot \frac{dx}{dt} + \frac{\partial q}{\partial y} \cdot \frac{dy}{dt} + \frac{\partial q}{\partial t}$$

$$= u \cdot \frac{\partial q}{\partial x} + v \cdot \frac{\partial q}{\partial y} + \frac{\partial q}{\partial t}. \tag{20}$$

Substituting from the partial differential equation equation 16, we obtain:

$$\frac{d}{dt}q(x(t), y(t), t) = 0.$$

Therefore, the scalar quantity $q$ is conserved along any Lagrangian path defined by the velocity field $\mathbf{v}(x, y, t)$. □

**Theorem A.2 (Constant-Delay Approximation for Advected Signals).** *Consider a spatial location $(x_1, y_1)$ lying on the streamline of a flow, with the streamline topology remaining unchanged. For a small time delay $\Delta t$, there exists an upstream point $(x_2, y_2)$ such that the scalar quantity $q$ at $(x_1, y_1)$ can be represented as its value at $(x_2, y_2)$ with a constant delay $\Delta t$ plus a small residual $\epsilon(t)$. That is,*

$$q(x_1, y_1, t) = q(x_2, y_2, t - \Delta t) + \epsilon(t). \tag{21}$$

*Proof.* For each $t$, let $\Gamma_\Delta(t)$ denote the spatial location at time $t - \Delta t$ that is carried by the flow to $(x_1, y_1)$ at time $t$. By scalar invariance along trajectories (Theorem A.1), we have the exact identity

$$q(x_1, y_1, t) = q\big(\Gamma_\Delta(t), \, t - \Delta t\big). \tag{22}$$

Let $(x_2, y_2)$ be an upstream point associated with this $\Delta t$ (e.g., the time average of $\Gamma_\Delta(t)$ over a short window), and define the time-dependent offset

$$\xi_\Delta(t) = \Gamma_\Delta(t) - (x_2, y_2). \tag{23}$$

Because the flow is steady and smooth, the speed bounded by a certain $U_{\max}$ and the upstream-point drift over a time interval $\Delta t$ is small. In particular, using the triangle inequality,

$$\|\xi_\Delta(t)\| \leq 2\, U_{\max}\, \Delta t. \tag{24}$$

Therefore, when $\Delta t$ is small, $\|\xi_\Delta(t)\|$ is also small. Applying a first-order Taylor expansion of $q$ with respect to the spatial variables at $\big((x_2, y_2),\, t - \Delta t\big)$ yields

$$q\big((x_2, y_2) + \xi_\Delta(t),\, t - \Delta t\big) = q(x_2, y_2,\, t - \Delta t) + \nabla q(x_2, y_2,\, t - \Delta t) \cdot \xi_\Delta(t) + \mathcal{O}\big(\|\xi_\Delta(t)\|^2\big). \quad (25)$$

Substituting this into equation 22 and defining

$$\eta(t) = \nabla q(x_2, y_2,\, t - \Delta t) \cdot \xi_\Delta(t) + \mathcal{O}\big(\|\xi_\Delta(t)\|^2\big), \quad (26)$$

we obtain

$$q(x_1, y_1, t) = q(x_2, y_2,\, t - \Delta t) + \eta(t). \quad (27)$$

This result shows that a downstream signal can be approximated by a time-shifted upstream signal plus a residual error $\eta(t)$. In practical scenarios, in addition to the time-shift residual $\eta(t)$ arising from variations in the advective speed, there also exists a *non-advective component* $r(t)$ caused by processes such as friction. Since both $\eta(t)$ and $r(t)$ represent deviations from the ideal constant-delay advection, we merge them into a single residual term $\epsilon(t)$. Thus, we obtain equation 21. $\qquad\square$

Figures 4a and 4b illustrate the constant-delay relation (Theorem A.2): wind speed series at locations separated by hundreds of kilometers but lying on the same streamline become nearly identical after applying an appropriate time shift. This example illustrates that advective pathways can be inferred from the background field, as regions with lag-aligned trends are likely to lie on the same streamline. We exploit this by aligning the target site's recent evolution with earlier segments from surrounding locations to trace upstream trajectories and forecast the target's subsequent trend.

## C  EXPERIMENTAL DETAILS

All methods are trained on eight Nvidia A6000 GPUs. During the training phase, the AdamW optimizer (Loshchilov & Hutter, 2019) was leveraged, accompanied by a weight decay parameter set to $0.05$.

MoP's hyperparameters conclude:

- **image_size**: Spatial grid size $[H, W]$.
- **patch_size**: Spatial patch size.
- **stride**: Spatial stride for patch splitting.
- **delta_t**: Lead time.
- **embed_dim**: Embedding dimension.
- **ctx_channels**: Background field input channels.
- **ts_channels**: Time series input channels.
- **ctx_length**: Background field input length.
- **ts_length**: Time series input length.
- **pred_len**: Length of forecast sequence.
- **search_block_num**: Number of retrieval–reasoning blocks in Advection Backtracker.
- **residual_block_num**: Number of self-attention fusion block in Residual Predictor.
- **lambda**: Weight of query-patch prediction loss.

For MoP, we tuned the task-sensitive hyperparameters (delta_t, patch_size, and stride). We chose the hyperparameters with the best validation performance and used this configuration for both in-domain and OOD evaluations. The detailed hyperparameter settings are shown in Table 4.

| Model: MoP | | |
|---|---|---|
| **Hyperparameter** | **4-h task** | **24-h task** |
| image_size | [32, 32] | [256, 256] |
| patch_size | [4, 4] | [48, 48] |
| stride | [1, 1] | [4, 4] |
| delta_t | 1 | 3 |
| embed_dim | 128 | 128 |
| ctx_channels | 1 | 1 |
| ts_channels | 1 | 1 |
| ctx_length | 4 | 24 |
| ts_length | 16 | 96 |
| pred_len | 16 | 96 |
| search_block_num | 1 | 2 |
| residual_block_num | 2 | 2 |
| lambda | 0.01 | 0.01 |

Table 4: Hyperparameters for MoP.

