# OpenReview forum: "Chasing the Wind: Background Flow Tracing for Wind Speed Forecasting"
_ICLR.cc/2026/Conference — ICLR 2026 Conference Withdrawn Submission_

### Official Review · Reviewer_Pv9z · 2025-10-20

**Soundness:** 2
**Presentation:** 3
**Contribution:** 2
**Rating:** 4
**Confidence:** 5

**Summary:**

This paper proposes MoP, a forecasting framework that models wind speed by retrieving upstream background field patches and refining them via residual correction. The authors justify this with an advection-based constant-delay assumption, and introduce a new dataset combining wind tower observations and ECMWF fields. The method achieves better long-horizon and cross-location generalization than existing time-series models like PatchTST, Crossformer, and context-fusion baselines like CrossViViT and FusionSF.

**Strengths:**

1. The model architecture is intuitively structured as a “retrieve upstream context + residual refinement” pipeline, making it relatively interpretable compared to black-box transformers.
2. The paper introduces a new dataset combining wind time series with background fields, which could be useful for future multimodal forecasting research.
3. Visualization of attention weights provides some transparency regarding how the model leverages spatial context.

**Weaknesses:**

1. The method is essentially retrieval-augmented time series forecasting, similar to existing matching or memory-based architectures.
2. The claimed OOD generalization advantage is not very convincing — traditional baselines already perform comparably in OOD settings, and the improvement margin of the proposed method is small.
3. The fusion between the two modalities (time series and wind field patches) appears relatively shallow and lacks theoretical or empirical justification. There is no clear ablation comparing “direct concatenation vs. MoP fusion structure.”
4. It is unclear whether the background wind field used by the model is realistically available in deployment scenarios. If it relies on reanalysis or numerical weather prediction products, then latency or cost could make real-time forecasting infeasible.
5. The paper frames the method as “physics-inspired,” but no hard physical constraints or PDE priors are actually enforced. The inductive bias remains soft, which raises the question of whether the gains truly come from physics or simply from more context.

**Questions:**

1. Is the required wind field context realistically obtainable in real-time, or does it rely on delayed or costly data sources?
2. Can MoP be used as a plug-in retrieval module inside a standard transformer forecaster, instead of a standalone architecture?
3. Would the method still provide benefits in domains without structured background fields (e.g., financial or load forecasting), or is it domain-specific?
4. Would the model perform similarly if the physical explanation were omitted and MoP were simply presented as a retrieval-augmented attention model?

---

### Official Review · Reviewer_p8Ua · 2025-10-31

**Soundness:** 2
**Presentation:** 3
**Contribution:** 2
**Rating:** 4
**Confidence:** 3

**Summary:**

This paper systematically reviews both physical models—based on numerical weather prediction and turbine dynamics—and data-driven approaches, including modern deep learning architectures such as CNNs, LSTMs, and Transformers, as well as their hybrid integrations.
The authors identify key challenges in WPF, including data quality and heterogeneity, limited spatial generalization across sites, model adaptability under dynamic weather conditions, and the need for multi-objective optimization balancing accuracy, stability, and interpretability.
The paper also presents a taxonomy of forecasting horizons and evaluation metrics, illustrating how modeling strategies differ by temporal scale.
Overall, it contributes a structured synthesis and roadmap for advancing WPF research, advocating for physics-informed learning, uncertainty modeling, multi-modal data fusion, and federated intelligence as critical directions toward more reliable and scalable wind power prediction.

**Strengths:**

S1.
The authors systematically integrate studies across diverse timescales, geographies, and modeling approaches, ensuring breadth and depth of coverage.

S2.
This paper is clear and well-organized.
It follows a logical progression from background principles to methodological categories, challenges, and future directions.

S3.
By consolidating fragmented research across physics, meteorology, and machine learning, the paper serves as a reference for new entrants to the field and a strategic roadmap for researchers pursuing hybrid or AI-enhanced WPF systems.

**Weaknesses:**

W1.
The paper introduces hybrid modeling (e.g., combining physical and data-driven approaches) as a promising direction but provides limited theoretical discussion on how these models bridge the gap between the physical and data-driven worlds.
For instance, the paper mentions that hybrid models can achieve better accuracy but lacks a detailed analysis on how these models integrate the complementary strengths of physics-based methods (e.g., understanding wind dynamics) and data-driven models (e.g., identifying complex patterns).

W2.
While the paper outlines the state of the art, it lacks empirical validation or case studies to illustrate how different forecasting approaches perform in real-world scenarios.

W3.
The paper identifies several key challenges in wind power forecasting, such as data heterogeneity and spatial generalization, but the discussion of potential solutions for these challenges is somewhat superficial.
For instance, while data quality issues are mentioned, the paper does not dive deeply into how the field could overcome these issues (e.g., through data augmentation, unsupervised pretraining, or cross-domain transfer learning).

W4.
While the paper proposes several exciting future research directions, such as physics-informed learning and federated learning, the discussion could be more focused on how these approaches can be practically implemented in the WPF domain.
For instance, the authors mention uncertainty quantification but do not elaborate on specific models or techniques that can be applied to this area.

**Questions:**

Q1.
You discuss hybrid models as a promising approach to bridge the gap between physical and data-driven methods, but how do you see the integration of physical principles (e.g., turbine aerodynamics or meteorological models) with neural network architectures (such as CNNs or LSTMs)?

Q2.
You mention data heterogeneity and spatial generalization as key challenges in wind power forecasting. How do you think transfer learning or domain adaptation methods can be employed to address these challenges, particularly when models trained on one wind farm or region must be adapted for another with different conditions (e.g., wind patterns, turbine models, etc.)?

---

### Official Review · Reviewer_tonh · 2025-10-31

**Soundness:** 2
**Presentation:** 2
**Contribution:** 2
**Rating:** 4
**Confidence:** 4

**Summary:**

This paper proposes MoP, a new framework for wind speed forecasting that uses background wind field data to model atmospheric advection. The key idea is a "constant-delay approximation" - the wind at a target location can be estimated from upstream winds a few hours earlier, plus a small correction. MoP has two main components: an Advection Backtracker that finds relevant upstream regions, and a Residual Predictor that adjusts for errors. The authors also introduce a new dataset combining local wind measurements with regional wind field data. Experiments show MoP outperforms existing methods, especially for 24-hour forecasts and at unseen locations, demonstrating its accuracy and strong generalization ability.

**Strengths:**

1. Strong and novel physical motivation.

2. Innovative and well-designed architecture.

3. Comprehensive Empirical Evaluation.

**Weaknesses:**

1. The core theorem relies on the assumption that "the streamline topology remains unchanged" over the forecast horizon. While the authors argue this is generally valid for large-scale flows over short periods, this assumption could be violated in regions with highly dynamic or complex weather patterns (e.g., near fronts or complex terrain). The paper would be strengthened by a discussion of the model's potential limitations or performance characteristics under such conditions.

2. Although the paper correctly notes that MoP is more efficient than modeling the full field evolution with neural ODEs, its computational cost compared to the other multi-modal baselines (e.g., CrossViViT, FusionSF) is not discussed. The iterative retrieval and matching process, while localized, could be non-trivial. A brief comparison of training/inference time or FLOPs would provide a more complete picture of the method's practicality.

**Questions:**

1. How sensitive is the model's performance to the quality and resolution of the background wind field? For instance, would the performance gains diminish if a lower-resolution or less accurate reanalysis product were used?

2. The $\delta_t$ hyperparameter is crucial. The ablation shows an optimal value of 3 hours for the 24h task. Was any analysis performed to see if this optimal $\delta_t$ is consistent across different geographic regions or weather regimes, or is it a dataset-specific tuning parameter?

3. The residual term $\epsilon(t)$ is meant to capture both temporal distortions and non-advective effects. Does the model provide any insight into the relative importance of these two sources of error? For example, can the magnitude of the predicted residual be correlated with specific meteorological conditions (e.g., high vs. low turbulence)?

---

### Official Review · Reviewer_fvJh · 2025-10-31

**Soundness:** 2
**Presentation:** 2
**Contribution:** 2
**Rating:** 4
**Confidence:** 4

**Summary:**

This paper focuses on the task of wind speed forecasting, which belongs to a type of multi-modal time series prediction that combines images (background wind fields) and time series (station observations). Inspired by the Constant-Delay Approximation for Advected Signals in fluid mechanics, the authors argue that local variations in wind speed do not evolve independently but are influenced by the momentum transport of upstream air masses.

Therefore, the paper proposes the MoP framework, which formulates wind speed forecasting as a spatiotemporal problem based on advective propagation. In terms of methodology, the model first constructs a cross-attention retrieval mechanism on spatiotemporal patches of the background field to identify upstream regions that exhibit time-delayed relationships with the target site, thereby achieving Advection Backtracking. Subsequently, the prediction results of the target patch are refined through the Residual Predictor to compensate for non-advective effects. For time series data, MoP concatenates the corresponding background field patches with the time series and performs prediction using an MLP.

**Strengths:**

1. This work possesses a good physical motivation, designing a reasonable Transformer block for the background field inspired by fluid mechanics theory.

2. The fluid mechanics approach initiates differential-based interactions among patches in the background field, which provides valuable insights for studies involving the flow of meteorological elements such as air quality and precipitation forecasting.

**Weaknesses:**

1) The authors claim that there is no public dataset containing both gridded background fields and wind speed observations, and they position the construction of such a dataset as one of their main research contributions. However, this claim is not accurate. The well-known SDWPF dataset [1], which was used in the KDD Cup 2022 and later published in Scientific Data, already provides exactly this type of data.

[1] SDWPF: A Dataset for Spatial Dynamic Wind Power Forecasting over a Large Turbine Array, 2024.


2) The model can be viewed as a modified ViT in its background field component, representing an innovation compared to Crossvivit and FusionSF. However, for the temporal channel, MoP adopts a concatenation and MLP approach, which is not equivalent to the previous Cross-Attention-based modal fusion mechanism and the Transformer method for time series. Given that the temporal patterns depicted in Figure2 are relatively simple, and considering that single-modal MLP and Mixer-based prediction methods in the Baseline section even outperform multimodal approaches in short-term predictions, we suspect that part of MoP's performance gain stems from its simple temporal feature extraction and cross-modal fusion approach. The authors should conduct additional supplementary experiments to rule out this possibility.

3) The authors should provide results of simple baseline methods such as Persistence, Mean, and Clear Sky, similar to those used in FusionSF, since these baselines demonstrate whether the prediction task of Solar Power or Wind Speed is inherently challenging. Although the authors introduce more advanced methods in the Related Work section—such as WindDragon and AirPhyHet—they do not include comparisons with them. Even coarse-grained meteorological forecasting models such as GraphCast and Pangu should be considered.

**Questions:**

see weakness

---

### Note · Authors · 2025-11-12

I have read and agree with the venue's withdrawal policy on behalf of myself and my co-authors.